# How much synthetic oxytocin is infused during labour? A review and analysis of regimens used in 12 countries

Deirdre Daly[1]*, Karin C. S. Minnie[2], Alwiena Blignaut[2], Ellen Blix[3], Anne Britt Vika Nilsen[4], Anna Dencker[5], Katrien Beeckman[6], Mechthild M. Gross[7], Jessica Pehlke-Milde[8], Susanne Grylka-Baeschlin[8], Martina Koenig-Bachmann[9], Jette Aaroe Clausen[10], Eleni Hadjigeorgiou[11], Sandra Morano[12], Laura Iannuzzi[13], Barbara Baranowska[14], Iwona Kiersnowska[15], Kerstin Uvnäs-Moberg[16]

1 School of Nursing and Midwifery, Trinity College Dublin, Dublin, Ireland, 2 NuMIQ research focus area: Research to promote quality of Nursing and Midwifery, North-West University, Potchefstroom, South Africa, 3 Faculty of Health Sciences, OsloMet—Oslo Metropolitan University, Oslo, Norway, 4 Department of Health and Caring Sciences, Western Norway University of Applied Sciences (HVL), Bergen, Norway, 5 Institute of Health and Care Sciences, Sahlgrenska Academy, University of Gothenburg, Gothenburg, Sweden, 6 Department of Public Health, Nursing and Midwifery Research group (NUMID), UZ Brussel, Vrije Universiteit Brussel; Midwifery Research Education and Policymaking (MidRep), University of Antwerp, Brussel, Belgium, 7 Midwifery Research and Education Unit, Hannover Medical School, Hannover, Germany, 8 Research Unit for Midwifery Science, Zurich University of Applied Sciences, Winterthur, Switzerland, 9 Health University of Applied Sciences Tyrol, Innsbruck, Austria, 10 Bachelor Degree Program in Midwifery, Copenhagen University College, Copenhagen, Denmark, 11 Nursing Department, Faculty of Health Science, Cyprus University of Technology, Limassol, Cyprus, 12 Department of Neurologic, Oculist, Gynaecologic, Maternal and Infant Sciences, University of Genoa, Genoa, Italy, 13 Department of Midwifery and Health Sciences, Faculty of Health and Social Sciences, Bournemouth University, Bournemouth, United Kingdom, 14 Department of Midwifery, Centre of Postgraduate Medical Education, Warsaw, Poland, 15 Department of Obstetrics and Perinatology, Medical University of Warsaw, Warsaw, Poland, 16 Swedish University of Agriculture, Skara, Sweden

* dalyd8@tcd.ie

**Data Availability Statement:** All relevant data are within the paper and its Supporting Information files

## Abstract

### Objective

To compare synthetic oxytocin infusion regimens used during labour, calculate the International Units (IU) escalation rate and total amount of IU infused over eight hours.

### Design

Observational study

### Setting

Twelve countries, eleven European and South Africa.

### Sample

National, regional or institutional-level regimens on oxytocin for induction and augmentation labour

**Funding:** This article is based upon work funded by the COST Action IS1405 BIRTH: "Building Intrapartum Research Through Health - An interdisciplinary whole system approach to understanding and contextualising physiological labour and birth" (http://www.cost.eu/COST_Actions/isch/IS1405), supported by EU COST (European Cooperation in Science and Technology).

**Competing interests:** The authors have declared that no competing interests exist.

**Abbreviations:** IU, International Unit; Hr, Hour; Min, Minutes; ml(s), Millilitre(s); mU(s), Milliunit (s); µg, Microgram; pg, Picogram.

## Methods

Data on oxytocin IU dose, infusion fluid amount, start dose, escalation rate and maximum dose were collected. Values for each regimen were converted to IU in 1000ml diluent. One IU corresponded to 1.67µg for doses provided in grams/micrograms. IU hourly dose increase rates were based on escalation frequency. Cumulative doses and total IU amount infused were calculated by adding the dose administered for each previous hour. Main Outcome Measures Oxytocin IU dose infused

## Results

Data were obtained on 21 regimens used in 12 countries. Details on the start dose, escalation interval, escalation rate and maximum dose infused were available from 16 regimens. Starting rates varied from 0.06 IU/hour to 0.90 IU/hour, and the maximum dose rate varied from 0.90 IU/hour to 3.60 IU/hour. The total amount of IU oxytocin infused, estimated over eight hours, ranged from 2.38 IU to 27.00 IU, a variation of 24.62 IU and an 11-fold difference.

## Conclusion

Current variations in oxytocin regimens for induction and augmentation of labour are inexplicable. It is crucial that the appropriate minimum infusion regimen is administered because synthetic oxytocin is a potentially harmful medication with serious consequences for women and babies when inappropriately used. Estimating the total amount of oxytocin IU received by labouring women, alongside the institution's mode of birth and neonatal outcomes, may deepen our understanding and be the way forward to identifying the optimal infusion regimen.

## Introduction

Infused synthetic oxytocin (Syntocinon®) [1] is one of the most widely used medications in labouring women. However, almost 70 years after it was first used in clinical practice [2] there is no agreement on the optimal infusion regimen to use during induction or augmentation of labour. Guideline recommendations vary [3–6] and debate about the risks and benefits of using high or low dose oxytocin regimens for induction of labour at term continues [7]. When labour is induced, discontinuing infused oxytocin after the onset of the active phase has no effect on the risk of having a caesarean section [8]. When labour onset is spontaneous, oxytocin administration for treating delayed progress, compared to delayed or no oxytocin administration, reduces the time to birth duration by approximately two hours but not the number of caesarean sections performed [9]. The authors concluded that a drug used for over 40 years to reduce the need for operative delivery was still not proven to be effective in its original primary role i.e., prevention of caesarean section [9]. While the effects of infused oxytocin on the uterus are not understood fully, one study found a decrease in the concentration of myometrial receptor binding sites and messenger RNA (mRNA), indicating that receptor desensitisation occurs [10]. A more recent study on levels of amniotic fluid lactate (AFL) just before augmentation with oxytocin found that low levels may aid the decision to continue augmentation but that high levels were predictive of an increased risk of caesarean section [11]. While more trials are

needed on the most effective dose, escalation rate and method of administration, pulsed or continuous, optimising the dose of oxytocin infused is important. The pulsatile pattern of oxytocin release, with increasing pulse frequency, observed during spontaneous labour is of physiologic significance [12, 13]: short lasting peaks start to occur at term and increase in frequency and size during labour, to a maximum of three pulses/10 minutes [14]. Infused oxytocin causes a flat pattern of release and a different pattern of uterine contractions to that seen during physiological labour. Contractions become irregular, more frequent, longer, and more painful, and may lead to hyperstimulation and compromised blood flow to the fetus. This abnormal contraction pattern increases the signalling in sensory nerves from the uterus, which leads to increased pain and stress levels in the mother [13].

When oxytocin is infused in doses up to 10 mU/min (0.6 IU/hour) plasma levels of oxytocin are within the range of oxytocin levels in physiological birth, around 40 picograms (pg)/ml. Oxytocin infusions influence the plasma levels of oxytocin in a dose dependent way and doubling the dose leads to doubling of the oxytocin levels. This means that when oxytocin is infused at a rate of 1.2 IU/hour the plasma oxytocin levels will be twice as high as during an infusion of 0.6 IU/hour or during normal physiological labour [12, 15, 16]. In addition, infused oxytocin does not cross the blood brain barrier and does not influence brain function in the same way as endogenous oxytocin [13]. Synthetic oxytocin is also a high-alert medication [17] that can have several adverse side effects [8, 9, 18], and it is implicated in obstetric claims in several countries [19–21]. Symptoms associated with over dose include uterine hyperstimulation and fetal heart rate changes [8, 9], meconium staining of the amniotic fluid, fetal asphyxia, placental abruption, amniotic fluid embolism and water intoxication [1]. Common maternal adverse effects, i.e., affecting 1 in 100 women, reported during the drug testing trials include headache, tachycardia or bradycardia and nausea and/or vomiting. Rare adverse effects, i.e., affecting 1 in 10,000 women, include anaphylactic reaction with dyspnoea, hypotension and skin rash [1]. Because of the impact on the physiological process of labour, and the plethora of potential side effects, we sought to estimate the total IU dose of synthetic oxytocin that would be infused over eight hours of labour, based on the start, escalating and maximum doses in available guidelines. This information is important in the context of over-medicalisation of normal pregnancy and birth [22] and because considerable proportions of labouring women worldwide receive infused oxytocin.

### Aim

To compare the guidelines on oxytocin infusion regimens used during labour in 12 countries and calculate the total amount of oxytocin IU received over an eight-hour period.

## Materials and methods

The study was conducted in 2017/18 as part of the COST Action IS1405 (BIRTH) (2014–2018) (https://www.cost.eu/actions/IS1405/BIRTH), a European-based network which aimed to advance scientific knowledge of the normal physiology of labour and birth. An email seeking expressions of interest in the review was sent to the 51 registered members of the Action's working group 4 who were based in 20 countries. To take part in the review, members needed knowledge of and access to their country's/region's/local institution's maternity services guideline on use of intrapartum oxytocin for induction and acceleration of labour. There were no exclusion criteria. A total of 18 members from 12 countries expressed an interest in taking part in the review.

A core group developed a data collection form based on the criteria used in the Irish clinical practice guideline on oxytocin to induce or accelerate labour [4]. The form was then sent to

the 18 participants for content validation and review. Subsequently, one item was included on the persons responsible for developing the guideline (obstetricians, midwives, users) and the final form contained 24 items (Table 1) (Available as a spreadsheet in S1 Data). Data collected included the oxytocin IU dose, type and amount of infusion fluid, start dose and escalation rate (ml/hour or IU/hour) (Table 1 and S1 Data).

## Ethics

As the study was a review of available national, regional and/or local regimens on intrapartum oxytocin administration, ethics committee approval was not required. Data collected included the country's name and type of guideline, national, regional or institution-level, and institution-level guidelines were identified as 'institution 1, 2 etc.' but not by name. No participant or individual-level data were collected. Written consent was not required. Participants volunteered to provide information requested. Eighteen members from 12 countries participated in the present study: 10 high-income and one upper-middle income European countries, and South Africa, a middle-income country. When members provided data on several guidelines or from several institutions in a country, we reported on the maximum and minimum regimens used in that country.

## Analysis

Data analysis broadly followed the approach used by Helbig et al [23]. Values for each regimen were converted to International Units (IU) in 1000ml of diluent, to obtain a common denominator for further dose calculations. One IU corresponded to 1.67μg for doses provided in grams or micrograms. The escalation rate (IU per hour) was calculated based on the frequency of escalation (escalation interval), being either 15, 20, 30 minutes. If millilitres (mls) were given for the rate increase, IU per hour was calculated by dividing the dose in the diluent (IU per 1000 mls) by 1000 (to obtain IU per ml) and then multiplied by the amount of mls escalated per hour. Using this escalation rate, the IU dose increase per hour could be calculated for each regimen, for a hypothetical eight-hour period, bearing in mind that escalation stopped when the maximum prescribed dose was reached or when effective uterine contractions were

**Table 1. Data collection form.**

| Country | Is there a national guideline? | Year | Oxytocin dose | Infusion fluid | Amount | Start dose | Escalation rate |
|---|---|---|---|---|---|---|---|
| Maximum dose | Is there a separate guideline for induction and acceleration (augmentation) of labour? | Is oxytocin titrated with uterine contractions? | How often are uterine contractions recorded? | Does guideline differ (dose, escalation rate, uterine activity, stage of labour (first or second stage)) for nulliparous women? | Does guideline differ (dose, escalation rate, uterine activity, stage of labour (first or second stage)) for multiparous women? | Does guideline differ (dose, escalation rate, uterine activity, stage of labour (first or second stage)) for women with a uterine scar? | Does guideline differ (dose, escalation rate, uterine activity, stage of labour (first or second stage)) for women with a twin pregnancy? |
| Does guideline differ (dose, escalation rate, uterine activity, stage of labour (first or second stage)) for women in pre-term labour? | Do local guidelines differ from national guidelines? (If yes, give example) | Variations in guidelines | Who can start oxytocin infusion? | Criteria for starting oxytocin infusion | Contraindication to use | Criteria for stopping oxytocin infusion | Who was involved in the development of the guideline? (Obstetricians, midwives, users) |

achieved. Cumulative doses were calculated by adding the dose administered for each hour to the doses administered over the previous hour/s, then the total dose of oxytocin IU infused over an eight-hour period was calculated. Calculations were performed by two authors (KM, AB), then checked and verified by another three authors (BB, IK, SG-B).

## Results

### Guideline content

Data collection forms on 21 regimens were received from 12 countries. A summary of regimens is presented in Table 2 and the detailed descriptions are in S2 Data.

Data on national guidelines were obtained from Denmark (for primiparous women only), Ireland, Norway, Sweden and South Africa (S2 Data). Data on two or more regimens, in use in different regions or institutions, were received from Austria, Belgium, Cyprus, Germany, Ireland, Poland and Switzerland (German and Italian-speaking regions and French-speaking region). When these regimens were different, we used data on the minimum and maximum regimens for calculations. Belgium's guidelines were from one institution and the national food and drug administration guidance but resulted in the same regimens. The Austrian data were from three institutions; the Irish data were from the national guideline and from one institution that used a different regimen; the Swiss data were from German and Italian-speaking regions and the French-speaking region. German data were from 35 university hospitals [23] and we included only data on the minimum and maximum regimens in this analysis. Apart from Denmark's regimen which was for augmentation of labour only, all other regimens were the same for both induction and augmentation of labour and almost all stated that the infusion should be titrated with uterine contractions.

### Criteria for use

Denmark's regimen was for primiparous women only, and Ireland's regimen differed for primiparous and multiparous women (S2 Data). Most regimens had different criteria on starting, stopping and contraindications. Belgium specified that oxytocin could only be used for induction of labour following administration of Prostin and a cervical Bishop Score ≥7. Five regimens stated the maximum desired uterine activity. In Ireland's national guideline, the rate of infusion should be reduced if contractions exceeded 7 in any 15-minute interval in primiparous women and 5 in 15 minutes in multiparous women. In Denmark, the rate of infusion should be reduced if the frequency of contractions exceeded 5 in 10 minutes in primiparous women, and Sweden's guideline stated that the infusion rate should be reduced if the frequency of contractions exceeded 5 per 10 minutes in both primiparous and multiparous women. Additional restrictions applied in certain regimens, indicating a maximum overall dose rather than a maximum dose rate. For example, the French-speaking region in Switzerland stopped escalation when the maximum overall dose of 10 IU was reached.

### Start dose, escalation interval, escalation rate and maximum dose rate

Based on the details provided, we were able to calculate the start dose, escalation interval, escalation rate and maximum dose rate increases of infused oxytocin on 16 regimens (Table 3). Starting doses ranged from 0.06 IU/hour in Ireland's and Poland's minimum regimens to 0.90 IU/hour in Germany's maximum regimen, and escalation intervals ranged from 15 to 60 minutes (Tables 4, 5, 6 and 7).

We were unable to perform calculations for Cyprus because variations in practices exist between institutions, and between obstetricians, within the private and public sectors. We also

Table 2. Oxytocin infusion regimens.

| Country | Is there a national guideline? | Year | Oxytocin dose | Infusion fluid | Amount | Start dose | Escalation rate | Maximum dose | Is there a separate guideline for induction and acceleration (augmentation) of labour? | Is oxytocin titrated with uterine contractions? | How often are uterine contractions recorded? |
|---|---|---|---|---|---|---|---|---|---|---|---|
| **Austria (Institution 1)** | No | - | 5 IU | 0.9% Saline or Ringer-Lactate | 500 mls | 9 mls/hour | 9 mls/hour, every 20 minutes | - | No | Yes | Continuously |
| **Austria (Institution 2)** | No | - | 5 IU | Electrolytic solution (e.g., ELO-MEL isoton or Ringer-Lactate "Fresenius") | 500 mls | 18 mls/hour | 20 minutes | 90 mls/hour | No | Yes | Continuous CTG |
| **Austria (Institution 3)** | No | - | 5 IU | Elo-mel | 500 mls | 18 mls/hour | 6 mls/hour | 120 mls/hour | Yes | Not stated | Continuous CTG |
| **Belgium (Institution)** | No | 2017 | 10 IU (1m IU = 2 drops (dr)) | Glucose 5% in Saline 0.45% | 1000 mls | 12 mls/ hours, 4 dr/ minute, 2 mU/minute | After 20 minutes, 24 mls/hour, 8 dr/ minute, 4mU; After 40 minutes, 36 mls/hour, 12 dr/minute, 6 mU; After 60 minutes, 54 mls/hour, 18 dr/minute, 9 mU; After 80 minutes, 84 mls/hour, 28 dr/minute, 4 mU; After 100 minutes, 120 mls/hour, 40 dr/minute, 20 mU; At 120 minutes, 180 mls/ hour, 60 dr/ minute, 30 mU (only on medical advice) | 120 mls/hour or 20 mU/ minute = 2 mls/ minute = 40 drops/minute | Yes | Yes | Continuous monitoring |

*(Continued)*

Table 2. (Continued)

| Country | Is there a national guideline? | Year | Oxytocin dose | Infusion fluid | Amount | Start dose | Escalation rate | Maximum dose | Is there a separate guideline for induction and acceleration (augmentation) of labour? | Is oxytocin titrated with uterine contractions? | How often are uterine contractions recorded? |
|---|---|---|---|---|---|---|---|---|---|---|---|
| **Belgium (Food and Drug Administration guidance)** | No | 2014 | 10 IU | Isotonic glucose solution | 1 IU/100 mls | 1 to 4 mU/minute = 0.1 to 0.4 mls/minute (2 to 8 dr/minute) | Progressive until desired effect and only if fetal heartrate and frequency and duration of the contractions are monitored closely. If hyperstimulation of the uterus or fetal distress occurs, the infusion should be stopped. If after infusion of 500 mls no regular contractions are observed, induction of labour should be stopped. | 120 ml/hour, or 20 mU/minute = 2 mls/minute = 40 drops/minute | No | Yes | Closely |
| **Cyprus (institution)** | No | 2016 | 2.5 IU or 5IU | Hartmans, Dextrose 5%. Saline 0.9% | 500 mls | Unclear | Unclear | Unclear | No | Yes | Early first stage every 15 minutes. Late first stage every 5–10 |
| **Denmark (National)** | Yes, on augmentation of labour only. | 2014 | 10 IU | 0.9% Saline | 1000 mls | 3.3 mU/minute (20 mls/hour) | 3.3 mU/minute (20 mls per hour) every 20 minutes | 30 mU/minute (180 mls/ hour) | Guideline on augmentation of labour only | Unclear. There is no mention of the amount of oxytocin infused per minute in relation to contractions. However, the guideline stresses that the infusion should only be escalated until there is a maximum of 5 contractions in a 10-minute interval | The guideline stresses that there should be no more than 5 contractions within a 10-minute interval but does not mention any systematic recording of uterine contractions |

*(Continued)*

**Table 2.** (Continued)

| Country | Is there a national guideline? | Year | Oxytocin dose | Infusion fluid | Amount | Start dose | Escalation rate | Maximum dose | Is there a separate guideline for induction and acceleration (augmentation) of labour? | Is oxytocin titrated with uterine contractions? | How often are uterine contractions recorded? |
|---|---|---|---|---|---|---|---|---|---|---|---|
| **Germany (institution) (minimum)** | No | - | 6 mU/ml | Ringer solution | 500 | 15 mU/minute | 15 mU/minute every 20 minutes | 60 mU/minute | Not stated | Yes | Continuous CTG |
| **Germany (Institution) (maximum)** | No | - | 60 mU | Saline 0.9% | 50 | 1 mU/minute | 1–4 mU/minute every 20 minutes | - | - | - | - |
| **Ireland (National and institution)** | Yes | 2016 | 10 IU | 0.9% Saline | 1000 mls | 1–5 mU/min (6–30 mls per hour) | 1–5 mU/minute (6–30 mls/hour) every 15–30 minutes | 30 mU/minute (180 mls/hour) | No | Yes | Measured over 15-minute intervals |
| **Italy (Institution** | No | 2018 | 5 IU | 0.9% Saline | 500 mls | 5 IU in 500 ml (10 mU/ml) Saline at 15 mls/hour | 15 mls/hour every 15 minutes, up to labour contractions of, normally, 3 strong and regular contractions lasting approx. 1 minute in 10 minutes) | 180 mls/hour | No | Yes | Continuously, via continuous monitoring EFM-CTG |
| **Norway (National)** | Yes | 2014 | 10 IU | 0.9% Saline or Ringer-Lactate | 1000 mls | 5 mU/minute (30 mls/hour) | 2.5 mU/minute every 15 minutes | 30 mU/minute | Not stated | Yes | Every 15–30 minutes, always before increasing dose |
| **Poland (Institution)** | No | 2017 | 5 IU/ml | 0.9% Saline, Glucose 5% | 50 mls | 5 IU in 50 mls Saline at 0.6–1.2 mls/hour | 1.2 mls/hour every 30 minutes, up to regular contractions (normally 3 strong and regular contractions lasting approx. 1 minute in 10 minutes | 18 mls/hour | No | Yes | Continuously, via continuous monitoring CTG |

(*Continued*)

Table 2. (Continued)

| Country | Is there a national guideline? | Year | Oxytocin dose | Infusion fluid | Amount | Start dose | Escalation rate | Maximum dose | Is there a separate guideline for induction and acceleration (augmentation) of labour? | Is oxytocin titrated with uterine contractions? | How often are uterine contractions recorded? |
|---|---|---|---|---|---|---|---|---|---|---|---|
| **South Africa (National)** | Yes | 2016 | 5 IU | Ringers Lactate | 1000 mls | 25 mls/hour (2U/minute) | 50 mls/hour every 30 minutes—hospital guidelines: double every 30 minutes: start with 3 mls for 30 minutes, then 6 mls/hour, then 12 mls/hour, then 24 mls/hour, then 48 mls/hour, then 60 mls/hour after 30 minutes. | 200 mls/hour or 3–4 contractions >40 seconds. If the infusion rate reaches 200 mls/hour and strong contractions are not achieved, increase the dose by starting an infusion of 10 units in 1000 mls at 150 mls/hour, increasing to 200 mls/hour if necessary. | No | Yes | Every 30 minutes for 10 minutes; continuous CTG, if available |
| **Sweden (National)** | Yes | 2011 | 5 IU (8.3 microgram/ml, 5 IU/ml)) | 0.9% Saline | 500 mls (or 2 mls/1000 mls, 10 IU/1000 mls), (i.e., same concentration but in 500 or 1000 mls) | 20 mls/ hour | Increase with 20 mls/hour every 20 minutes | 180 mls/hour (can be raised after that but after doctor's order only) | No | Yes | Measured over 10 minutes. There is no mention of how often contractions should be recorded but it states that there should also be a special checklist for oxytocin and that the contractions should be noted every time the oxytocin dose is changed and if not changed, every hour. |

*(Continued)*

**Table 2.** (Continued)

| Country | Is there a national guideline? | Year | Oxytocin dose | Infusion fluid | Amount | Start dose | Escalation rate | Maximum dose | Is there a separate guideline for induction and acceleration (augmentation) of labour? | Is oxytocin titrated with uterine contractions? | How often are uterine contractions recorded? |
|---|---|---|---|---|---|---|---|---|---|---|---|
| **Switzerland French speaking regions** | No | 2012 | 5 IU | 0.9% Saline | 500 mls | 30 mls/hour | 15 mls/hour every 30 minutes. If there are >7 contractions in 15 minutes then no further escalation | 360 mls/hour. Maximum total dose: 10 IU | No | Yes | Measured over 15-minute intervals |
| **Switzerland German and Italian speaking regions** | No | 2012 | 5 IU | Ringerfundin Braun | 500 mls | 12 mls/hour | 12 mls/hour every 30 minutes. If there are 3–4 contractions in 10 minutes, no further escalation; if there are >4 contractions in 10 minutes, reduce by 12 mls/hour | 108 mls/hour (0.018 IU/min), to be titrated for a maximum of 30 minutes | No | Yes | Measured over 10-minute intervals |

Table 3. Oxytocin infusion start dose, escalation interval, escalation rate and maximum dose rate.

| Country | Regimen source | Regimen type | Start Dose Rate (mU/ hr) | Escalation Interval (Minutes) | Escalation Rate (IU/hr) | Maximum Dose Rate (IU/hr) |
|---|---|---|---|---|---|---|
| Ireland | National | Minimum | 60 | 30 | 0.06 | 1.80 |
| | Institution | Maximum | 300 | 15 | 0.30 | 1.80 |
| Italy | Institution | - | 150 | 15 | 0.15 | 1.80 |
| Cyprus | Institution | Minimum | 175 | 60 | 0.18 | 0.90 |
| | Institution | Maximum | 350 | 60 | 0.35 | 1.80 |
| Germany | Institution | Minimum | 72 | 60 | 0.07 | 0.43 |
| | Institution | Maximum | 900 | 20 | 0.90 | 3.60 |
| Switzerland | German & Italian speaking regions | - | 120 | 30 | 0.12 | 1.08 |
| | French speaking region | - | 150 | 30 | 0.15 | 3.60 |
| South Africa | National | - | 125 | 30 | 0.25 | 1.00 |
| Norway | National | - | 300 | 15 | 0.15 | 1.80 |
| Denmark | National | - | 200 | 20 | 0.20 | 1.80 |
| Sweden | National | - | 200 | 20 | 0.20 | 1.80 |
| Belgium | Institution (no fixed escalation rate)* | - | 120 | 20 | 0.12 | 1.20 |
| | | | | 20 | 0.12 | 1.20 |
| | | | | 20 | 0.18 | 1.20 |
| | | | | 20 | 0.30 | 1.20 |
| | | | | 20 | 0.36 | 1.20 |
| | | | | 20 | 0.60 | 1.20 |
| Belgium | Food and Drug Administration guidance** | - | 120 | 20 | - | 1.20 |
| Austria | Institution | Minimum | 90 | 20 | 0.09 | 0.90 |
| | Institution | Maximum | 180 | 20 | 0.06 | 1.20 |
| Poland | Institution | Minimum | 60 | 30 | 0.12 | 1.80 |
| | Institution | Maximum | 120 | 30 | 0.12 | 1.80 |

*No fixed escalation rate (IU/hr); after 120 minutes, seek medical advice before further escalation.

**See Table 2

calculated the total dose of oxytocin that would be infused if the infusion was escalated up to a certain point (i.e., sufficient uterine contractions) and then maintained over an eight-hour period, as a hypothetical labour duration. Calculations for the regimens that would lead to the lowest and highest infused doses, Germany's minimum and maximum regimens, are shown in Tables 8 and 9 calculations for all regimens are presented as supporting information (S3 Data).

In the minimum regimen, the total dose infused would be 2.38 IU if the infusion was escalated at each time point up to eight hours. In the maximum regimen, the total dose infused at one hour and 20 minutes would be 2.70 IU if the infusion was increased for one hour and then maintained, and the total dose infused over an eight-hour period would be 20.70 IU. If the infusion was escalated for two hours and then maintained, the total dose infused over eight hours would be 27.00 IU. In Ireland's minimum regimen, if the infusion was increased for one hour and then maintained for a further seven hours, the total dose infused would be 0.93 IU. If the infusion was escalated at each interval, the total dose infused over eight hours would be 4.08 IU, more than four times higher, but three times lower than the 13.28 IU that would be infused in the maximum

**Table 4. Regimens with 15-minute escalation interval.**

| Escalation interval (15 minutes) | 00:00 | 00:15 | 00:30 | 00:45 | 01:00 | 01:15 | 01:30 | 01:45 | 02:00 | 02:15 | 02:30 | 02:45 | 03:00 | 03:15 | 03:30 | 03:45 | 04:00 | 04:15 | 04:30 | 04:45 | 05:00 | 05:15 | 05:30 | 05:45 | 06:00 | 06:15 | 06:30 | 06:45 | 07:00 | 07:15 | 07:30 | 07:45 | 08:00 |
|---|---|---|---|---|---|---|---|---|---|---|---|---|---|---|---|---|---|---|---|---|---|---|---|---|---|---|---|---|---|---|---|---|---|
| **Ireland (maximum)** | | | | | | | | | | | | | | | | | | | | | | | | | | | | | | | | | |
| Rate Escalation (IU/Hr) | 0.30 | 0.30 | 0.30 | 0.30 | 0.30 | 0.30 | 0.00 | 0.00 | 0.00 | 0.00 | 0.00 | 0.00 | 0.00 | 0.00 | 0.00 | 0.00 | 0.00 | 0.00 | 0.00 | 0.00 | 0.00 | 0.00 | 0.00 | 0.00 | 0.00 | 0.00 | 0.00 | 0.00 | 0.00 | 0.00 | 0.00 | 0.00 | 0.00 |
| Dose Rate (IU/Hr) | 0.30 | 0.60 | 0.90 | 1.20 | 1.50 | 1.80 | 1.80 | 1.80 | 1.80 | 1.80 | 1.80 | 1.80 | 1.80 | 1.80 | 1.80 | 1.80 | 1.80 | 1.80 | 1.80 | 1.80 | 1.80 | 1.80 | 1.80 | 1.80 | 1.80 | 1.80 | 1.80 | 1.80 | 1.80 | 1.80 | 1.80 | 1.80 | 1.80 |
| Dose (IU) given at each time interval | --- | 0.08 | 0.15 | 0.23 | 0.30 | 0.38 | 0.45 | 0.45 | 0.45 | 0.45 | 0.45 | 0.45 | 0.45 | 0.45 | 0.45 | 0.45 | 0.45 | 0.45 | 0.45 | 0.45 | 0.45 | 0.45 | 0.45 | 0.45 | 0.45 | 0.45 | 0.45 | 0.45 | 0.45 | 0.45 | 0.45 | 0.45 | 0.45 |
| Total Dose (IU) infused | --- | 0.08 | 0.23 | 0.45 | 0.75 | 1.13 | 1.58 | 2.03 | 2.48 | 2.93 | 3.38 | 3.83 | 4.28 | 4.73 | 5.18 | 5.63 | 6.08 | 6.53 | 6.98 | 7.43 | 7.88 | 8.33 | 8.78 | 9.23 | 9.68 | 10.13 | 10.58 | 11.03 | 11.48 | 11.93 | 12.38 | 12.83 | 13.28 |
| **Italy** | | | | | | | | | | | | | | | | | | | | | | | | | | | | | | | | | |
| Rate Escalation (IU/Hr) | 0.15 | 0.15 | 0.15 | 0.15 | 0.15 | 0.15 | 0.15 | 0.15 | 0.15 | 0.15 | 0.15 | 0.15 | 0.00 | 0.00 | 0.00 | 0.00 | 0.00 | 0.00 | 0.00 | 0.00 | 0.00 | 0.00 | 0.00 | 0.00 | 0.00 | 0.00 | 0.00 | 0.00 | 0.00 | 0.00 | 0.00 | 0.00 | 0.00 |
| Dose Rate (IU/Hr) | 0.15 | 0.30 | 0.45 | 0.60 | 0.75 | 0.90 | 1.05 | 1.20 | 1.35 | 1.50 | 1.65 | 1.80 | 1.80 | 1.80 | 1.80 | 1.80 | 1.80 | 1.80 | 1.80 | 1.80 | 1.80 | 1.80 | 1.80 | 1.80 | 1.80 | 1.80 | 1.80 | 1.80 | 1.80 | 1.80 | 1.80 | 1.80 | 1.80 |
| Dose (IU) given at each time interval | --- | 0.04 | 0.08 | 0.11 | 0.15 | 0.19 | 0.23 | 0.26 | 0.30 | 0.34 | 0.38 | 0.41 | 0.45 | 0.45 | 0.45 | 0.45 | 0.45 | 0.45 | 0.45 | 0.45 | 0.45 | 0.45 | 0.45 | 0.45 | 0.45 | 0.45 | 0.45 | 0.45 | 0.45 | 0.45 | 0.45 | 0.45 | 0.45 |
| Total Dose (IU) infused | --- | 0.04 | 0.11 | 0.23 | 0.38 | 0.56 | 0.79 | 1.05 | 1.35 | 1.69 | 2.06 | 2.48 | 2.93 | 3.38 | 3.83 | 4.28 | 4.73 | 5.18 | 5.63 | 6.08 | 6.53 | 6.98 | 7.43 | 7.88 | 8.33 | 8.78 | 9.23 | 9.68 | 10.13 | 10.58 | 11.03 | 11.48 | 11.93 |
| **Norway** | | | | | | | | | | | | | | | | | | | | | | | | | | | | | | | | | |
| Rate Escalation (IU/Hr) | 0.30 | 0.15 | 0.15 | 0.15 | 0.15 | 0.15 | 0.15 | 0.15 | 0.15 | 0.15 | 0.15 | 0.00 | 0.00 | 0.00 | 0.00 | 0.00 | 0.00 | 0.00 | 0.00 | 0.00 | 0.00 | 0.00 | 0.00 | 0.00 | 0.00 | 0.00 | 0.00 | 0.00 | 0.00 | 0.00 | 0.00 | 0.00 | 0.00 |
| Dose Rate (IU/Hr) | 0.30 | 0.45 | 0.60 | 0.75 | 0.90 | 1.05 | 1.20 | 1.35 | 1.50 | 1.65 | 1.80 | 1.80 | 1.80 | 1.80 | 1.80 | 1.80 | 1.80 | 1.80 | 1.80 | 1.80 | 1.80 | 1.80 | 1.80 | 1.80 | 1.80 | 1.80 | 1.80 | 1.80 | 1.80 | 1.80 | 1.80 | 1.80 | 1.80 |
| Dose (IU) given at each time interval | --- | 0.08 | 0.11 | 0.15 | 0.19 | 0.23 | 0.26 | 0.30 | 0.34 | 0.38 | 0.41 | 0.45 | 0.45 | 0.45 | 0.45 | 0.45 | 0.45 | 0.45 | 0.45 | 0.45 | 0.45 | 0.45 | 0.45 | 0.45 | 0.45 | 0.45 | 0.45 | 0.45 | 0.45 | 0.45 | 0.45 | 0.45 | 0.45 |
| Total Dose (IU) infused | --- | 0.08 | 0.19 | 0.34 | 0.53 | 0.75 | 1.01 | 1.31 | 1.65 | 2.03 | 2.44 | 2.89 | 3.34 | 3.79 | 4.24 | 4.69 | 5.14 | 5.59 | 6.04 | 6.49 | 6.94 | 7.39 | 7.84 | 8.29 | 8.74 | 9.19 | 9.64 | 10.09 | 10.54 | 10.99 | 11.44 | 11.89 | 12.34 |

**Table 5. Regimens with 20-minute escalation interval.**

| Escalation interval (20 minutes) | 00:00 | 00:20 | 00:40 | 01:00 | 01:20 | 01:40 | 02:00 | 02:20 | 02:40 | 03:00 | 03:20 | 03:40 | 04:00 | 04:20 | 04:40 | 05:00 | 05:20 | 05:40 | 06:00 | 06:20 | 06:40 | 07:00 | 07:20 | 07:40 | 08:00 |
|---|---|---|---|---|---|---|---|---|---|---|---|---|---|---|---|---|---|---|---|---|---|---|---|---|---|
| **German (maximum)** | | | | | | | | | | | | | | | | | | | | | | | | | |
| Rate Escalation (IU/hr) | --- | 0.90 | 0.90 | 0.90 | 0.00 | 0.00 | 0.00 | 0.00 | 0.00 | 0.00 | 0.00 | 0.00 | 0.00 | 0.00 | 0.00 | 0.00 | 0.00 | 0.00 | 0.00 | 0.00 | 0.00 | 0.00 | 0.00 | 0.00 | 0.00 |
| Dose Rate (IU/hr) | 0.90 | 1.80 | 2.70 | 3.60 | 3.60 | 3.60 | 3.60 | 3.60 | 3.60 | 3.60 | 3.60 | 3.60 | 3.60 | 3.60 | 3.60 | 3.60 | 3.60 | 3.60 | 3.60 | 3.60 | 3.60 | 3.60 | 3.60 | 3.60 | 3.60 |
| Dose (IU) given at each time interval | --- | 0.30 | 0.60 | 0.90 | 1.20 | 1.20 | 1.20 | 1.20 | 1.20 | 1.20 | 1.20 | 1.20 | 1.20 | 1.20 | 1.20 | 1.20 | 1.20 | 1.20 | 1.20 | 1.20 | 1.20 | 1.20 | 1.20 | 1.20 | 1.20 |
| Total Dose (IU) infused | --- | 0.30 | 0.90 | 1.80 | 3.00 | 4.20 | 5.40 | 6.60 | 7.80 | 9.00 | 10.20 | 11.40 | 12.60 | 13.80 | 15.00 | 16.20 | 17.40 | 18.60 | 19.80 | 21.00 | 22.20 | 23.40 | 24.60 | 25.80 | 27.00 |
| **Denmark** | | | | | | | | | | | | | | | | | | | | | | | | | |
| Rate Escalation (IU/hr) | --- | 0.20 | 0.20 | 0.20 | 0.20 | 0.20 | 0.20 | 0.20 | 0.20 | 0.00 | 0.00 | 0.00 | 0.00 | 0.00 | 0.00 | 0.00 | 0.00 | 0.00 | 0.00 | 0.00 | 0.00 | 0.00 | 0.00 | 0.00 | 0.00 |
| Dose Rate (IU/hr) | 0.20 | 0.40 | 0.60 | 0.80 | 1.00 | 1.20 | 1.40 | 1.60 | 1.80 | 1.80 | 1.80 | 1.80 | 1.80 | 1.80 | 1.80 | 1.80 | 1.80 | 1.80 | 1.80 | 1.80 | 1.80 | 1.80 | 1.80 | 1.80 | 1.80 |
| Dose (IU) given at each time interval | --- | 0.07 | 0.13 | 0.20 | 0.27 | 0.33 | 0.40 | 0.47 | 0.53 | 0.60 | 0.60 | 0.60 | 0.60 | 0.60 | 0.60 | 0.60 | 0.60 | 0.60 | 0.60 | 0.60 | 0.60 | 0.60 | 0.60 | 0.60 | 0.60 |
| Total Dose (IU) infused | --- | 0.07 | 0.20 | 0.40 | 0.67 | 1.00 | 1.40 | 1.87 | 2.40 | 3.00 | 3.60 | 4.20 | 4.80 | 5.40 | 6.00 | 6.60 | 7.20 | 7.80 | 8.40 | 9.00 | 9.60 | 10.20 | 10.80 | 11.40 | 12.00 |
| **Sweden** | | | | | | | | | | | | | | | | | | | | | | | | | |
| Rate Escalation (IU/hr) | --- | 0.20 | 0.20 | 0.20 | 0.20 | 0.20 | 0.20 | 0.20 | 0.20 | 0.00 | 0.00 | 0.00 | 0.00 | 0.00 | 0.00 | 0.00 | 0.00 | 0.00 | 0.00 | 0.00 | 0.00 | 0.00 | 0.00 | 0.00 | 0.00 |
| Dose Rate (IU/hr) | 0.20 | 0.40 | 0.60 | 0.80 | 1.00 | 1.20 | 1.40 | 1.60 | 1.80 | 1.80 | 1.80 | 1.80 | 1.80 | 1.80 | 1.80 | 1.80 | 1.80 | 1.80 | 1.80 | 1.80 | 1.80 | 1.80 | 1.80 | 1.80 | 1.80 |
| Dose (IU) given at each time interval | --- | 0.07 | 0.13 | 0.20 | 0.27 | 0.33 | 0.40 | 0.47 | 0.53 | 0.60 | 0.60 | 0.60 | 0.60 | 0.60 | 0.60 | 0.60 | 0.60 | 0.60 | 0.60 | 0.60 | 0.60 | 0.60 | 0.60 | 0.60 | 0.60 |
| Total Dose (IU) infused | --- | 0.07 | 0.20 | 0.40 | 0.67 | 1.00 | 1.40 | 1.87 | 2.40 | 3.00 | 3.60 | 4.20 | 4.80 | 5.40 | 6.00 | 6.60 | 7.20 | 7.80 | 8.40 | 9.00 | 9.60 | 10.20 | 10.80 | 11.40 | 12.00 |
| **Belgium** | | | | | | | | | | | | | | | | | | | | | | | | | |
| Rate Escalation (IU/hr) | --- | 0.12 | 0.12 | 0.18 | 0.30 | 0.36 | 0.00 | 0.00 | 0.00 | 0.00 | 0.00 | 0.00 | 0.00 | 0.00 | 0.00 | 0.00 | 0.00 | 0.00 | 0.00 | 0.00 | 0.00 | 0.00 | 0.00 | 0.00 | 0.00 |
| Dose Rate (IU/hr) | 0.12 | 0.24 | 0.36 | 0.54 | 0.84 | 1.20 | 1.20 | 1.20 | 1.20 | 1.20 | 1.20 | 1.20 | 1.20 | 1.20 | 1.20 | 1.20 | 1.20 | 1.20 | 1.20 | 1.20 | 1.20 | 1.20 | 1.20 | 1.20 | 1.20 |
| Dose (IU) given at each time interval | --- | 0.04 | 0.08 | 0.12 | 0.18 | 0.28 | 0.40 | 0.40 | 0.40 | 0.40 | 0.40 | 0.40 | 0.40 | 0.40 | 0.40 | 0.40 | 0.40 | 0.40 | 0.40 | 0.40 | 0.40 | 0.40 | 0.40 | 0.40 | 0.40 |
| Total Dose (IU) infused | --- | 0.04 | 0.12 | 0.24 | 0.42 | 0.70 | 1.10 | 1.50 | 1.90 | 2.30 | 2.70 | 3.10 | 3.50 | 3.90 | 4.30 | 4.70 | 5.10 | 5.50 | 5.90 | 6.30 | 6.70 | 7.10 | 7.50 | 7.90 | 8.30 |
| **Austria (minimum)** | | | | | | | | | | | | | | | | | | | | | | | | | |
| Rate Escalation (IU/hr) | --- | 0.09 | 0.09 | 0.09 | 0.09 | 0.09 | 0.09 | 0.09 | 0.09 | 0.09 | 0.00 | 0.00 | 0.00 | 0.00 | 0.00 | 0.00 | 0.00 | 0.00 | 0.00 | 0.00 | 0.00 | 0.00 | 0.00 | 0.00 | 0.00 |
| Dose Rate (IU/hr) | 0.09 | 0.18 | 0.27 | 0.36 | 0.45 | 0.54 | 0.63 | 0.72 | 0.81 | 0.90 | 0.90 | 0.90 | 0.90 | 0.90 | 0.90 | 0.90 | 0.90 | 0.90 | 0.90 | 0.90 | 0.90 | 0.90 | 0.90 | 0.90 | 0.90 |
| Dose (IU) given at each time interval | --- | 0.03 | 0.06 | 0.09 | 0.12 | 0.15 | 0.18 | 0.21 | 0.24 | 0.27 | 0.30 | 0.30 | 0.30 | 0.30 | 0.30 | 0.30 | 0.30 | 0.30 | 0.30 | 0.30 | 0.30 | 0.30 | 0.30 | 0.30 | 0.30 |
| Total Dose (IU) infused | --- | 0.03 | 0.09 | 0.18 | 0.30 | 0.45 | 0.63 | 0.84 | 1.08 | 1.35 | 1.65 | 1.95 | 2.25 | 2.55 | 2.85 | 3.15 | 3.45 | 3.75 | 4.05 | 4.35 | 4.65 | 4.95 | 5.25 | 5.55 | 5.85 |
| **Austria (maximum)** | | | | | | | | | | | | | | | | | | | | | | | | | |
| Rate Escalation (IU/hr) | --- | 0.06 | 0.06 | 0.06 | 0.06 | 0.06 | 0.06 | 0.06 | 0.06 | 0.06 | 0.06 | 0.06 | 0.06 | 0.06 | 0.06 | 0.06 | 0.06 | 0.06 | 0.00 | 0.00 | 0.00 | 0.00 | 0.00 | 0.00 | 0.00 |
| Dose Rate (IU/hr) | 0.18 | 0.24 | 0.30 | 0.36 | 0.42 | 0.48 | 0.54 | 0.60 | 0.66 | 0.72 | 0.78 | 0.84 | 0.90 | 0.96 | 1.02 | 1.08 | 1.14 | 1.20 | 1.20 | 1.20 | 1.20 | 1.20 | 1.20 | 1.20 | 1.20 |
| Dose (IU) given at each time interval | --- | 0.06 | 0.08 | 0.10 | 0.12 | 0.14 | 0.16 | 0.18 | 0.20 | 0.22 | 0.24 | 0.26 | 0.28 | 0.30 | 0.32 | 0.34 | 0.36 | 0.38 | 0.40 | 0.40 | 0.40 | 0.40 | 0.40 | 0.40 | 0.40 |
| Total Dose (IU) infused | --- | 0.06 | 0.14 | 0.24 | 0.36 | 0.50 | 0.66 | 0.84 | 1.04 | 1.26 | 1.50 | 1.76 | 2.04 | 2.34 | 2.66 | 3.00 | 3.36 | 3.74 | 4.14 | 4.54 | 4.94 | 5.34 | 5.74 | 6.14 | 6.54 |

**Table 6. Regimens with 30-minute escalation interval.**

| Escalation interval (30 minutes) | 00:00 | 00:30 | 01:00 | 01:30 | 02:00 | 02:30 | 03:00 | 03:30 | 04:00 | 04:30 | 05:00 | 05:30 | 06:00 | 06:30 | 07:00 | 07:30 | 08:00 |
|---|---|---|---|---|---|---|---|---|---|---|---|---|---|---|---|---|---|
| **Ireland (minimum)** | | | | | | | | | | | | | | | | | |
| Rate Escalation *(IU/hr)* | --- | 0.06 | 0.06 | 0.06 | 0.06 | 0.06 | 0.06 | 0.06 | 0.06 | 0.06 | 0.06 | 0.06 | 0.06 | 0.06 | 0.06 | 0.06 | 0.06 |
| Dose Rate *(IU/hr)* | 0.06 | 0.12 | 0.18 | 0.24 | 0.30 | 0.36 | 0.42 | 0.48 | 0.54 | 0.60 | 0.66 | 0.72 | 0.78 | 0.84 | 0.90 | 0.96 | 1.02 |
| Dose *(IU) given at each time interval* | --- | 0.03 | 0.06 | 0.09 | 0.12 | 0.15 | 0.18 | 0.21 | 0.24 | 0.27 | 0.30 | 0.33 | 0.36 | 0.39 | 0.42 | 0.45 | 0.48 |
| Total Dose *(IU) infused* | --- | 0.03 | 0.09 | 0.18 | 0.30 | 0.45 | 0.63 | 0.84 | 1.08 | 1.35 | 1.65 | 1.98 | 2.34 | 2.73 | 3.15 | 3.60 | 4.08 |
| **Switzerland (German & Italian Regions)** | | | | | | | | | | | | | | | | | |
| Rate Escalation *(IU/hr)* | --- | 0.12 | 0.12 | 0.12 | 0.12 | 0.12 | 0.12 | 0.12 | 0.12 | 0.00 | 0.00 | 0.00 | 0.00 | 0.00 | 0.00 | 0.00 | 0.00 |
| Dose Rate *(IU/hr)* | 0.12 | 0.24 | 0.36 | 0.48 | 0.60 | 0.72 | 0.84 | 0.96 | 1.08 | 1.08 | 1.08 | 1.08 | 1.08 | 1.08 | 1.08 | 1.08 | 1.08 |
| Dose *(IU) given at each time interval* | --- | 0.06 | 0.12 | 0.18 | 0.24 | 0.30 | 0.36 | 0.42 | 0.48 | 0.54 | 0.54 | 0.54 | 0.54 | 0.54 | 0.54 | 0.54 | 0.54 |
| Total Dose *(IU) infused* | --- | 0.06 | 0.18 | 0.36 | 0.60 | 0.90 | 1.26 | 1.68 | 2.16 | 2.70 | 3.24 | 3.78 | 4.32 | 4.86 | 5.40 | 5.94 | 6.48 |
| **Switzerland (French Region)** | | | | | | | | | | | | | | | | | |
| Rate Escalation *(IU/hr)* | --- | 0.15 | 0.15 | 0.15 | 0.15 | 0.15 | 0.15 | 0.15 | 0.15 | 0.15 | 0.15 | 0.15 | 0.15 | 0.15 | 0.15 | 0.15 | 0.15 |
| Dose Rate *(IU/hr)* | 0.15 | 0.30 | 0.45 | 0.60 | 0.75 | 0.90 | 1.05 | 1.20 | 1.35 | 1.50 | 1.65 | 1.80 | 1.95 | 2.10 | 2.25 | 2.40 | 2.55 |
| Dose *(IU) given at each time interval* | --- | 0.08 | 0.15 | 0.23 | 0.30 | 0.38 | 0.45 | 0.53 | 0.60 | 0.68 | 0.75 | 0.83 | 0.90 | 0.98 | 1.05 | 1.13 | 1.20 |
| Total Dose *(IU) infused* | --- | 0.08 | 0.23 | 0.45 | 0.75 | 1.13 | 1.58 | 2.10 | 2.70 | 3.38 | 4.13 | 4.95 | 5.85 | 6.83 | 7.88 | 9.00 | 10.20 |
| **South Africa** | | | | | | | | | | | | | | | | | |
| Rate Escalation *(IU/hr)* | --- | 0.25 | 0.25 | 0.25 | 0.12 | 0.00 | 0.00 | 0.00 | 0.00 | 0.00 | 0.00 | 0.00 | 0.00 | 0.00 | 0.00 | 0.00 | 0.00 |
| Dose Rate *(IU/hr)* | 0.13 | 0.38 | 0.63 | 0.88 | 1.00 | 1.00 | 1.00 | 1.00 | 1.00 | 1.00 | 1.00 | 1.00 | 1.00 | 1.00 | 1.00 | 1.00 | 1.00 |
| Dose *(IU) given at each time interval* | --- | 0.06 | 0.19 | 0.31 | 0.44 | 0.50 | 0.50 | 0.50 | 0.50 | 0.50 | 0.50 | 0.50 | 0.50 | 0.50 | 0.50 | 0.50 | 0.50 |
| Total Dose *(IU) infused* | --- | 0.06 | 0.25 | 0.56 | 1.00 | 1.50 | 2.00 | 2.49 | 2.99 | 3.49 | 3.99 | 4.48 | 4.98 | 5.48 | 5.98 | 6.47 | 6.97 |
| **Poland (minimum)** | | | | | | | | | | | | | | | | | |
| Rate Escalation *(IU/hr)* | --- | 0.12 | 0.12 | 0.12 | 0.12 | 0.12 | 0.12 | 0.12 | 0.12 | 0.12 | 0.12 | 0.12 | 0.12 | 0.12 | 0.12 | 0.12 | 0.12 |
| Dose Rate *(IU/hr)* | 0.06 | 0.18 | 0.30 | 0.42 | 0.54 | 0.66 | 0.78 | 0.90 | 1.02 | 1.14 | 1.26 | 1.38 | 1.50 | 1.62 | 1.74 | 1.80 | 1.80 |
| Dose *(IU) given at each time interval* | --- | 0.03 | 0.09 | 0.15 | 0.21 | 0.27 | 0.33 | 0.39 | 0.45 | 0.51 | 0.57 | 0.63 | 0.69 | 0.75 | 0.81 | 0.87 | 0.90 |
| Total Dose *(IU) infused* | --- | 0.03 | 0.12 | 0.27 | 0.48 | 0.75 | 1.08 | 1.47 | 1.92 | 2.43 | 3.00 | 3.63 | 4.32 | 5.07 | 5.88 | 6.75 | 7.65 |
| **Poland (maximum)** | | | | | | | | | | | | | | | | | |
| Rate Escalation *(IU/hr)* | --- | 0.12 | 0.12 | 0.12 | 0.12 | 0.12 | 0.12 | 0.12 | 0.12 | 0.12 | 0.12 | 0.12 | 0.12 | 0.12 | 0.12 | 0.00 | 0.00 |
| Dose Rate *(IU/hr)* | 0.12 | 0.24 | 0.36 | 0.48 | 0.60 | 0.72 | 0.84 | 0.96 | 1.08 | 1.20 | 1.32 | 1.44 | 1.56 | 1.68 | 1.80 | 1.80 | 1.80 |
| Dose *(IU) given at each time interval* | --- | 0.06 | 0.12 | 0.18 | 0.24 | 0.30 | 0.36 | 0.42 | 0.48 | 0.54 | 0.60 | 0.66 | 0.72 | 0.78 | 0.84 | 0.90 | 0.90 |
| Total Dose *(IU) infused* | --- | 0.06 | 0.18 | 0.36 | 0.60 | 0.90 | 1.26 | 1.68 | 2.16 | 2.70 | 3.30 | 3.96 | 4.68 | 5.46 | 6.30 | 7.20 | 8.10 |

regimen (S3 Data). The differences between the total doses infused in Poland's and Austria's minimum and maximum regimens would be much lower, 0.45 IU and 0.69 IU respectively.

The time when the maximum oxytocin infusion dose would be reached, assuming escalation continued according to the regimen, is shown in Table 10.

## Total amount of oxytocin IU infused

The total amount of oxytocin IU that would be infused over this hypothetical eight-hour was highest in Germany's maximum regimen (27.00 IU) and lowest in Germany's minimum regimen (2.38 IU), and the variation was 24.62 IU (Table 10). Fig 2 shows the variations in cumulative dose of oxytocin infused over eight hours, by regimen and country.

## Discussion

Findings show considerable differences in the oxytocin regimens and escalation dose rates recommended in the 16 regimens reviewed, resulting in considerable variation in the total

Table 7. Regimen with 60-minute escalation interval.

| Escalation interval (60 minutes) | 00:00 | 00:20 | 00:40 | 01:00 | 01:20 | 01:40 | 02:00 | 02:20 | 02:40 | 03:00 | 03:20 | 03:40 | 04:00 | 04:20 | 04:40 | 05:00 | 05:20 | 05:40 | 06:00 | 06:20 | 06:40 | 07:00 | 07:20 | 07:40 | 08:00 |
|---|---|---|---|---|---|---|---|---|---|---|---|---|---|---|---|---|---|---|---|---|---|---|---|---|---|
| **Germany (minimum)** | | | | | | | | | | | | | | | | | | | | | | | | | |
| Rate Escalation (IU/hr) | --- | 0.00 | 0.00 | 0.07 | 0.00 | 0.00 | 0.07 | 0.00 | 0.00 | 0.07 | 0.00 | 0.00 | 0.07 | 0.00 | 0.00 | 0.07 | 0.00 | 0.00 | 0.00 | 0.00 | 0.00 | 0.00 | 0.00 | 0.00 | 0.00 |
| Dose Rate (IU/hr) | 0.07 | 0.07 | 0.07 | 0.14 | 0.14 | 0.14 | 0.22 | 0.22 | 0.22 | 0.29 | 0.29 | 0.29 | 0.36 | 0.36 | 0.36 | 0.43 | 0.43 | 0.43 | 0.43 | 0.43 | 0.43 | 0.43 | 0.43 | 0.43 | 0.43 |
| Dose (IU) given at each time interval | --- | 0.02 | 0.02 | 0.02 | 0.05 | 0.05 | 0.05 | 0.07 | 0.07 | 0.07 | 0.10 | 0.10 | 0.10 | 0.12 | 0.12 | 0.12 | 0.14 | 0.14 | 0.14 | 0.14 | 0.14 | 0.14 | 0.14 | 0.14 | 0.14 |
| Total Dose (IU) infused | --- | 0.02 | 0.05 | 0.07 | 0.12 | 0.17 | 0.22 | 0.29 | 0.36 | 0.43 | 0.53 | 0.62 | 0.72 | 0.84 | 0.96 | 1.08 | 1.22 | 1.37 | 1.51 | 1.66 | 1.80 | 1.94 | 2.09 | 2.23 | 2.38 |

**Table 8. Germany minimum regimen—oxytocin (IU) dose infused when plateaued at indicated hour.**

| Germany (minimum) | 00:00 | 00:20 | 00:40 | 01:00 | 01:20 | 01:40 | 02:00 | 02:20 | 02:40 | 03:00 | 03:20 | 03:40 | 04:00 | 04:20 | 04:40 | 05:00 | 05:20 | 05:40 | 06:00 | 06:20 | 06:40 | 07:00 | 07:20 | 07:40 | 08:00 |
|---|---|---|---|---|---|---|---|---|---|---|---|---|---|---|---|---|---|---|---|---|---|---|---|---|---|
| Rate Escalation (IU/hr) | --- | 0.00 | 0.00 | 0.07 | 0.00 | 0.00 | 0.07 | 0.00 | 0.00 | 0.07 | 0.00 | 0.00 | 0.07 | 0.00 | 0.00 | 0.07 | 0.00 | 0.00 | 0.00 | 0.00 | 0.00 | 0.00 | 0.00 | 0.00 | 0.00 |
| Dose Rate (IU/hr) | 0.07 | 0.07 | 0.07 | 0.14 | 0.14 | 0.14 | 0.22 | 0.22 | 0.22 | 0.29 | 0.29 | 0.29 | 0.36 | 0.36 | 0.36 | 0.43 | 0.43 | 0.43 | 0.43 | 0.43 | 0.43 | 0.43 | 0.43 | 0.43 | 0.43 |
| Dose (IU) given at each time interval | --- | 0.02 | 0.02 | 0.02 | 0.05 | 0.05 | 0.05 | 0.07 | 0.07 | 0.07 | 0.10 | 0.10 | 0.10 | 0.12 | 0.12 | 0.12 | 0.14 | 0.14 | 0.14 | 0.14 | 0.14 | 0.14 | 0.14 | 0.14 | 0.14 |
| Total Dose (IU) infused | --- | 0.02 | 0.05 | 0.07 | 0.12 | 0.17 | 0.22 | 0.29 | 0.36 | 0.43 | 0.53 | 0.62 | 0.72 | 0.84 | 0.96 | 1.08 | 1.22 | 1.37 | 1.51 | 1.66 | 1.80 | 1.94 | 2.09 | 2.23 | 2.38 |
| Dose (IU) infused if plateaus after indicated hour | | | | | | | | | | | | | | | | | | | | | | | | | |
| 1 hour | --- | 0.02 | 0.05 | 0.07 | 0.10 | 0.12 | 0.14 | 0.17 | 0.19 | 0.22 | 0.24 | 0.26 | 0.29 | 0.31 | 0.34 | 0.36 | 0.38 | 0.41 | 0.43 | 0.46 | 0.48 | 0.50 | 0.53 | 0.55 | 0.58 |
| 2 hours | --- | 0.02 | 0.05 | 0.07 | 0.12 | 0.17 | 0.22 | 0.26 | 0.31 | 0.36 | 0.41 | 0.46 | 0.50 | 0.55 | 0.60 | 0.65 | 0.70 | 0.74 | 0.79 | 0.84 | 0.89 | 0.94 | 0.98 | 1.03 | 1.08 |
| 3 hours | --- | 0.02 | 0.05 | 0.07 | 0.12 | 0.17 | 0.22 | 0.29 | 0.36 | 0.43 | 0.50 | 0.58 | 0.65 | 0.72 | 0.79 | 0.86 | 0.94 | 1.01 | 1.08 | 1.15 | 1.22 | 1.30 | 1.37 | 1.44 | 1.51 |
| 4 hours | --- | 0.02 | 0.05 | 0.07 | 0.12 | 0.17 | 0.22 | 0.29 | 0.36 | 0.43 | 0.53 | 0.62 | 0.72 | 0.82 | 0.91 | 1.01 | 1.10 | 1.20 | 1.30 | 1.39 | 1.49 | 1.58 | 1.68 | 1.78 | 1.87 |
| 5 hours | --- | 0.02 | 0.05 | 0.07 | 0.12 | 0.17 | 0.22 | 0.29 | 0.36 | 0.43 | 0.53 | 0.62 | 0.72 | 0.84 | 0.96 | 1.08 | 1.22 | 1.37 | 1.51 | 1.66 | 1.80 | 1.94 | 2.09 | 2.23 | 2.38 |
| 6 hours | --- | 0.02 | 0.05 | 0.07 | 0.12 | 0.17 | 0.22 | 0.29 | 0.36 | 0.43 | 0.53 | 0.62 | 0.72 | 0.84 | 0.96 | 1.08 | 1.22 | 1.37 | 1.51 | 1.66 | 1.80 | 1.94 | 2.09 | 2.23 | 2.38 |
| 7 hours | --- | 0.02 | 0.05 | 0.07 | 0.12 | 0.17 | 0.22 | 0.29 | 0.36 | 0.43 | 0.53 | 0.62 | 0.72 | 0.84 | 0.96 | 1.08 | 1.22 | 1.37 | 1.51 | 1.66 | 1.80 | 1.94 | 2.09 | 2.23 | 2.38 |
| 8 hours | --- | 0.02 | 0.05 | 0.07 | 0.12 | 0.17 | 0.22 | 0.29 | 0.36 | 0.43 | 0.53 | 0.62 | 0.72 | 0.84 | 0.96 | 1.08 | 1.22 | 1.37 | 1.51 | 1.66 | 1.80 | 1.94 | 2.09 | 2.23 | 2.38 |

Table 9. Germany maximum regimen—oxytocin (IU) dose infused when plateaued at indicated hour.

| Germany (maximum) | 00:00 | 00:20 | 00:40 | 01:00 | 01:20 | 01:40 | 02:00 | 02:20 | 02:40 | 03:00 | 03:20 | 03:40 | 04:00 | 04:20 | 04:40 | 05:00 | 05:20 | 05:40 | 06:00 | 06:20 | 06:40 | 07:00 | 07:20 | 07:40 | 08:00 |
|---|---|---|---|---|---|---|---|---|---|---|---|---|---|---|---|---|---|---|---|---|---|---|---|---|---|
| Rate Escalation (IU/hr) | --- | 0.90 | 0.90 | 0.90 | 0.00 | 0.00 | 0.00 | 0.00 | 0.00 | 0.00 | 0.00 | 0.00 | 0.00 | 0.00 | 0.00 | 0.00 | 0.00 | 0.00 | 0.00 | 0.00 | 0.00 | 0.00 | 0.00 | 0.00 | 0.00 |
| Dose Rate (IU/hr) | 0.90 | 1.80 | 2.70 | 3.60 | 3.60 | 3.60 | 3.60 | 3.60 | 3.60 | 3.60 | 3.60 | 3.60 | 3.60 | 3.60 | 3.60 | 3.60 | 3.60 | 3.60 | 3.60 | 3.60 | 3.60 | 3.60 | 3.60 | 3.60 | 3.60 |
| Dose (IU) given at each time interval | --- | 0.30 | 0.60 | 0.90 | 1.20 | 1.20 | 1.20 | 1.20 | 1.20 | 1.20 | 1.20 | 1.20 | 1.20 | 1.20 | 1.20 | 1.20 | 1.20 | 1.20 | 1.20 | 1.20 | 1.20 | 1.20 | 1.20 | 1.20 | 1.20 |
| Total Dose (IU) infused | --- | 0.30 | 0.90 | 1.80 | 3.00 | 4.20 | 5.40 | 6.60 | 7.80 | 9.00 | 10.20 | 11.40 | 12.60 | 13.80 | 15.00 | 16.20 | 17.40 | 18.60 | 19.80 | 21.00 | 22.20 | 23.40 | 24.60 | 25.80 | 27.00 |
| Dose (IU) infused if plateaus after indicated hour | | | | | | | | | | | | | | | | | | | | | | | | | |
| 1 hour | --- | 0.30 | 0.90 | 1.80 | 2.70 | 3.60 | 4.50 | 5.40 | 6.30 | 7.20 | 8.10 | 9.00 | 9.90 | 10.80 | 11.70 | 12.60 | 13.50 | 14.40 | 15.30 | 16.20 | 17.10 | 18.00 | 18.90 | 19.80 | 20.70 |
| 2 hours | --- | 0.30 | 0.90 | 1.80 | 3.00 | 4.20 | 5.40 | 6.60 | 7.80 | 9.00 | 10.20 | 11.40 | 12.60 | 13.80 | 15.00 | 16.20 | 17.40 | 18.60 | 19.80 | 21.00 | 22.20 | 23.40 | 24.60 | 25.80 | 27.00 |
| 3 hours | --- | 0.30 | 0.90 | 1.80 | 3.00 | 4.20 | 5.40 | 6.60 | 7.80 | 9.00 | 10.20 | 11.40 | 12.60 | 13.80 | 15.00 | 16.20 | 17.40 | 18.60 | 19.80 | 21.00 | 22.20 | 23.40 | 24.60 | 25.80 | 27.00 |
| 4 hours | --- | 0.30 | 0.90 | 1.80 | 3.00 | 4.20 | 5.40 | 6.60 | 7.80 | 9.00 | 10.20 | 11.40 | 12.60 | 13.80 | 15.00 | 16.20 | 17.40 | 18.60 | 19.80 | 21.00 | 22.20 | 23.40 | 24.60 | 25.80 | 27.00 |
| 5 hours | --- | 0.30 | 0.90 | 1.80 | 3.00 | 4.20 | 5.40 | 6.60 | 7.80 | 9.00 | 10.20 | 11.40 | 12.60 | 13.80 | 15.00 | 16.20 | 17.40 | 18.60 | 19.80 | 21.00 | 22.20 | 23.40 | 24.60 | 25.80 | 27.00 |
| 6 hours | --- | 0.30 | 0.90 | 1.80 | 3.00 | 4.20 | 5.40 | 6.60 | 7.80 | 9.00 | 10.20 | 11.40 | 12.60 | 13.80 | 15.00 | 16.20 | 17.40 | 18.60 | 19.80 | 21.00 | 22.20 | 23.40 | 24.60 | 25.80 | 27.00 |
| 7 hours | --- | 0.30 | 0.90 | 1.80 | 3.00 | 4.20 | 5.40 | 6.60 | 7.80 | 9.00 | 10.20 | 11.40 | 12.60 | 13.80 | 15.00 | 16.20 | 17.40 | 18.60 | 19.80 | 21.00 | 22.20 | 23.40 | 24.60 | 25.80 | 27.00 |
| 8 hours | --- | 0.30 | 0.90 | 1.80 | 3.00 | 4.20 | 5.40 | 6.60 | 7.80 | 9.00 | 10.20 | 11.40 | 12.60 | 13.80 | 15.00 | 16.20 | 17.40 | 18.60 | 19.80 | 21.00 | 22.20 | 23.40 | 24.60 | 25.80 | 27.00 |

Table 10. Time when maximum oxytocin infusion dose would be reached with maximum escalation.

| *No further escalation after this hour* | Time when maximum oxytocin infusion dose (IU) is reached | | | | | | | |
|---|---|---|---|---|---|---|---|---|
| | **1** | **2** | **3** | **4** | **5** | **6** | **7** | **8** |
| **Ireland (minimum)** | 0.93 | 1.74 | 2.43 | 3.00 | 3.45 | 3.78 | 3.99 | 4.08 |
| **Ireland (maximum)** | 9.15 | 13.28 | 13.28 | 13.28 | 13.28 | 13.28 | 13.28 | 13.28 |
| **Italy** | 4.58 | 8.55 | 11.93 | 11.93 | 11.93 | 11.93 | 11.93 | 11.93 |
| **Germany (minimum)** | 0.58 | 1.08 | 1.51 | 1.87 | 2.38 | 2.38 | 2.38 | 2.38 |
| **Germany (maximum)** | 20.70 | 27.00 | 27.00 | 27.00 | 27.00 | 27.00 | 27.00 | 27.00 |
| **Norway** | 5.78 | 9.75 | 12.34 | 12.34 | 12.34 | 12.34 | 12.34 | 12.34 |
| **Denmark** | 4.60 | 8.60 | 12.00 | 12.00 | 12.00 | 12.00 | 12.00 | 12.00 |
| **Sweden** | 4.60 | 8.60 | 12.00 | 12.00 | 12.00 | 12.00 | 12.00 | 12.00 |
| **Belgium** | 2.76 | 8.30 | 8.30 | 8.30 | 8.30 | 8.30 | 8.30 | 8.30 |
| **Austria (minimum)** | 2.07 | 3.87 | 5.82 | 5.85 | 5.85 | 5.85 | 5.85 | 5.85 |
| **Austria (maximum)** | 2.34 | 3.54 | 4.56 | 5.40 | 6.06 | 6.54 | 6.54 | 6.54 |
| **Switzerland (German & Italian speaking Regions)** | 1.86 | 3.48 | 4.86 | 6.00 | 6.48 | 6.48 | 6.48 | 6.48 |
| **Switzerland (French speaking Regions)** | 2.33 | 4.35 | 6.08 | 7.50 | 8.63 | 9.45 | 9.98 | 10.00 |
| **South Africa** | 2.88 | 6.25 | 6.97 | 6.97 | 6.97 | 6.97 | 6.97 | 6.97 |
| **Poland (minimum)** | 1.38 | 3.00 | 4.38 | 5.52 | 6.42 | 7.08 | 7.50 | 7.65 |
| **Poland (maximum)** | 1.86 | 3.48 | 4.86 | 6.00 | 6.90 | 7.56 | 7.98 | 8.10 |

IU/hour dose rates increases plateaued at two hours in three regimens, at three hours in five regimens and between four and eight hours in the remaining regimens. The IU/hour dose rate increases had not plateaued at eight hours in four regimens. The dose rate increase over an eight-hour period is presented graphically in Fig 1.

amount of oxytocin IU that would be infused over eight hours of labour. The total amount of oxytocin that would be infused was estimated at 27.00 IU in Germany's maximum regimen, 11

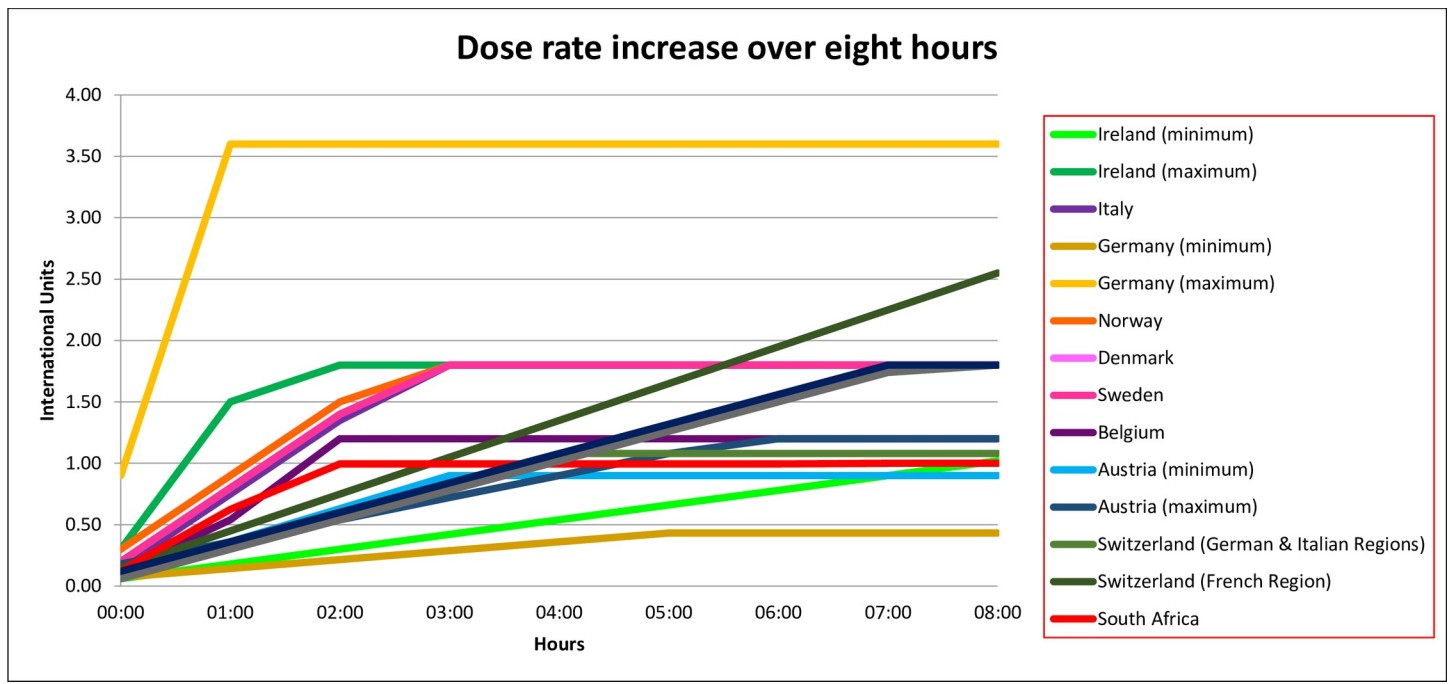

**Fig 1. Dose rate increase over an eight-hour period.**

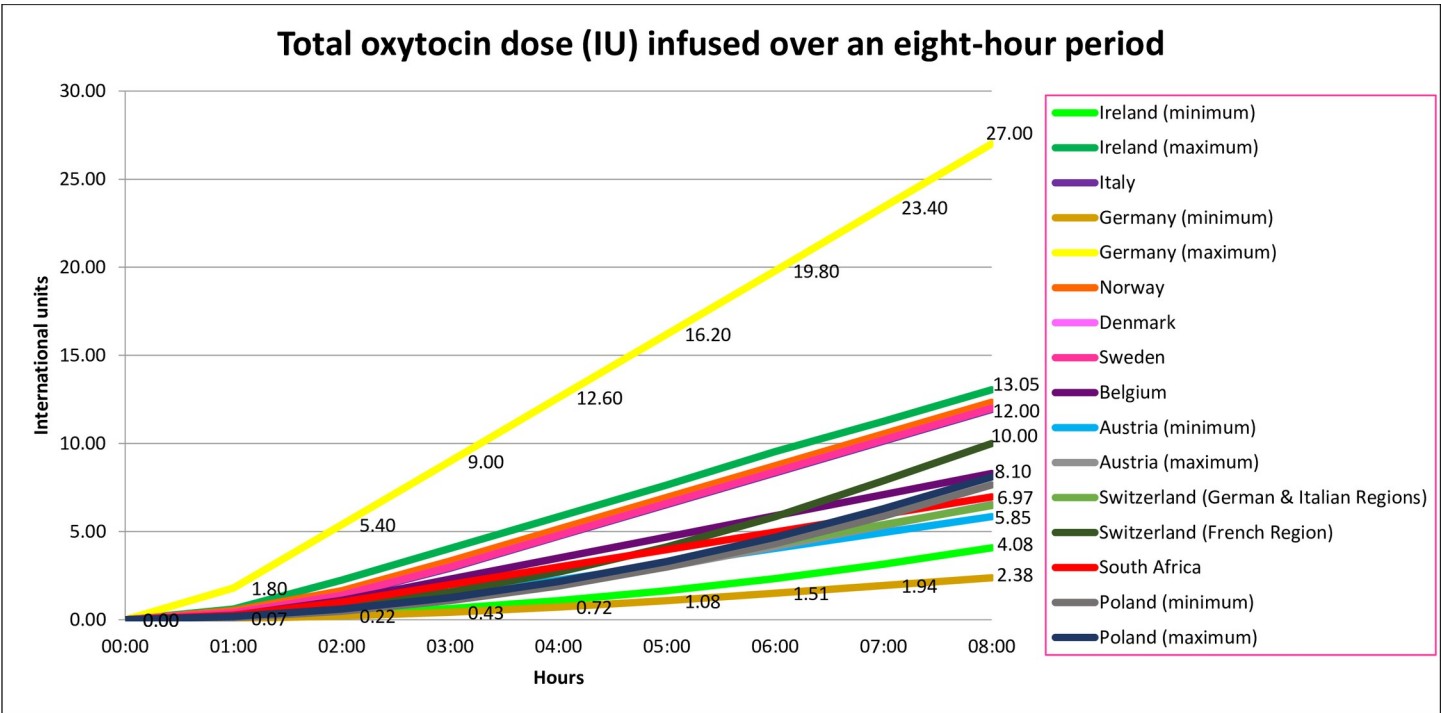

**Fig 2. Total oxytocin dose (IU) infused over an eight-hour period.**

times higher than the 2.38 IU infused in their minimum regimen. In Ireland's maximum regimen the total amount of oxytocin infused after eight hours was 13.28 IU, more than three times higher than in their minimum regimen (4.08 IU). The total amount of oxytocin IU infused in the other regimens ranged from 5.85 IU to 13.05 IU. Three regimens stipulated that escalation of doses should cease when the desired number of uterine contractions in a specified time period was reached, therefore it is possible that maximum doses are seldom, if ever, reached in practice. Due to rate escalation differences, huge variations in the total dose infused after eight hours were noted. However, it is important to note that dose escalation is continued/discontinued on clinical need, thus extreme outliers in dose administration will probably be very rare. Of critical importance is that these regimens show a gradual increase in oxytocin levels, even in the early stages of escalation, and lack the physiologically significant pulses of oxytocin observed at increasing frequency during spontaneous labour [12]. While there is no agreement on the most effective regimen or administration dose, it is reasonable to suggest that the lowest possible effective amount should be administered to achieve the desired outcome and avoid or minimise side effects related to the amount and speed of oxytocin administered. This is vital in the context of the number of women worldwide who undergo induction or augmentation of labour, irrespective of the indication, and receive intrapartum synthetic oxytocin infusions, and current concern regarding over-medicalisation of normal labour and birth [22].

Previous papers have reported variations in oxytocin regimens [24–27] and concluded that a standardised administration regimen is needed. Our study shows that differences in regimens persist, even within countries, and these differences result in considerable variations in the total amount of oxytocin IU administered. Such variations in the total amount infused raise questions about the over-administration in some settings, especially if lower doses are effective and when considered alongside worldwide rates of induction and acceleration of labour. For

example, one study demonstrated that women spent 408 minutes on intravenous oxytocin following induction, and 321 minutes following augmentation, of labour [28]. They also found that intravenous oxytocin titration was not recorded in 53% of the records reviewed. The proportion of women whose labour was induced ranged from 22% in the United States of America (USA) (2006) [3] and France (2016) [29] to 29.1% in New South Wales (NSW) (2007) [30] and 71.0% in Iran (2011–12) [22]. Proportions ranged from 6.8% in Lithuania to 33.0% in Wallonia in 2010 [31] and from 18.6% to 45.7% in primiparous women across the 19 maternity hospitals/units in Ireland in 2009 [32]. While the proportion of women whose labour was augmented with oxytocin declined from 53% in 2001–02 to 26% 10 years later (2011–12) in the USA [33], 78.9% of women in India underwent labour augmentation with oxytocin in 2011 [22]. This latter proportion is, according to the authors, concerning and strongly suggestive of the routine over-medicalisation of normal pregnancy and birth [22]. While synthetic oxytocin is a useful medication, it is also dangerous when over or inappropriately used [34]. Short-term intrapartum consequences of over or inappropriate use include myometrial hyperstimulation (excessive frequency of contractions), hypertonicity (increased uterine tone between contractions), fetal heart rate changes, fetal hypoxia etc. [1], and an analysis of litigation cases in Norway [20], Sweden [35] and the UK [19, 34] showed that the use of oxytocin played a role in many cases. This financial burden to health services is of obvious importance but of concern are the consequences for women, both physical and psychological. Research on the longer term effects of synthetic oxytocin, during the postpartum period and beyond, is emerging and some studies suggest that intrapartum synthetic oxytocin may alter the endogenous oxytocin system and influence women's stress, mood and behaviour [36]. At two months postpartum, women who were breastfeeding exclusively had received significantly less infused synthetic oxytocin during labour, compared with women who were not breastfeeding exclusively, and higher doses were associated with greater depressive, anxious, and somatic symptoms [37]. Even within the first year postpartum, exposure to peripartum synthetic oxytocin increased women's risk of being diagnosed with a postpartum depressive or anxiety disorder or being prescribed an antidepressant or anxiolytic medication [38]. Two of these studies did not mention the dose of oxytocin administered but the average amount administered in the third study (36 IU) far exceeded the regimens presented in our study. These three studies [36, 37, 38] did not differentiate between oxytocin administered intrapartum or postpartum and different types of administration are likely to give rise to different effects. In addition, the authors did not mention co-administration of epidural analgesia and other medical interventions which may be associated with the effects caused by infused oxytocin. While the results of these studies must be interpreted with caution, a plausible explanation for increased levels of anxiety and depression is that the increased pain and stress experienced by women who received high amounts of synthetic oxytocin led to higher stress levels, including higher levels of cortisol, and that a sustained increase in stress levels was induced by infused oxytocin [13]. Nevertheless, it follows that it is preferable to infuse oxytocin at the lowest rate possible in order to avoid hyperstimulation. In light of number of women who receive infused oxytocin during labour, and the considerable variation in the total amount of IU infused, it is vital that women and their babies are exposed to the minimum dose of a medication that alters the physiological processes and has significant short and long-term consequences. It is also reasonable to assume that the variations in regimens seen are, like other intrapartum interventions in high income countries (HICs) [39], being driven by clinical practice patterns rather than medical indications.

## Strengths and limitations

The strengths of this study lie in the comparison of 16 oxytocin infusion regimens sourced from 12 countries. The study is innovative in that it moves beyond the regimens used to

estimating the total amount of oxytocin IU that would be infused over an 8-hour period, and results show an 11-fold difference between minimum and maximum amounts.

However, our study has some limitations. First, data on several regimens were taken directly from written protocols, national, regional or institutional, and transcribed into the data collection form but some were not, and there was no way of verifying data that were provided by these other means. Second, estimating the total amount of oxytocin IU infused by calculating start dose and escalation dose rates from regimens ignores the obvious effect of titration with uterine contractions. However, results show the wide variation in the total amounts of oxytocin IU infused when maximum rates are reached and maintained over defined time periods.

## Conclusion

In the era of evidence-based health care, the fact that such widespread variation exists in the use of infused oxytocin, and in the total amount infused, reflects potential overuse in many settings. All maternity care professionals are driven by the need to reduce avoidable maternal and neonatal morbidity and mortality, and it is crucial that intrapartum interventions designed to reduce risk for some who have complications are not used routinely for others who are healthy. It is also crucial that the total IU amount of a medication with known harmful side effects is documented. Estimating the total amount of oxytocin IU received by women who enter labour at term gestation, either spontaneously or following induction, documenting titration rates and the institution's mode of birth and neonatal outcomes may deepen our understanding and be the way forward.

## Supporting information

**S1 Data. Review of oxytocin regimens—data collection form.**
(XLSX)

**S2 Data. Oxytocin regimens.**
(XLSX)

**S3 Data. Start dose, escalation interval, escalation rate, maximum dose.**
(XLSX)

## Acknowledgments

This study is part of the EU COST Action IS1405 "Building Intrapartum Research Through Health—An interdisciplinary whole system approach to understanding and contextualising physiological labour and birth".

## Author Contributions

**Conceptualization:** Deirdre Daly, Kerstin Uvnäs-Moberg.

**Data curation:** Deirdre Daly, Karin C. S. Minnie, Ellen Blix, Anne Britt Vika Nilsen, Anna Dencker, Katrien Beeckman, Mechthild M. Gross, Jessica Pehlke-Milde, Susanne Grylka-Baeschlin, Martina Koenig-Bachmann, Jette Aaroe Clausen, Eleni Hadjigeorgiou, Laura Iannuzzi, Barbara Baranowska, Iwona Kiersnowska.

**Formal analysis:** Deirdre Daly, Karin C. S. Minnie, Alwiena Blignaut, Susanne Grylka-Baeschlin, Barbara Baranowska, Iwona Kiersnowska.

**Investigation:** Deirdre Daly.

**Methodology:** Deirdre Daly, Karin C. S. Minnie.

**Project administration:** Deirdre Daly.

**Resources:** Deirdre Daly.

**Supervision:** Deirdre Daly.

**Validation:** Deirdre Daly, Karin C. S. Minnie, Kerstin Uvnäs-Moberg.

**Writing – original draft:** Deirdre Daly, Karin C. S. Minnie, Ellen Blix, Katrien Beeckman, Mechthild M. Gross, Jessica Pehlke-Milde, Kerstin Uvnäs-Moberg.

**Writing – review & editing:** Deirdre Daly, Karin C. S. Minnie, Alwiena Blignaut, Anne Britt Vika Nilsen, Anna Dencker, Susanne Grylka-Baeschlin, Martina Koenig-Bachmann, Jette Aaroe Clausen, Eleni Hadjigeorgiou, Sandra Morano, Laura Iannuzzi, Barbara Baranowska, Iwona Kiersnowska, Kerstin Uvnäs-Moberg.

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
