## [Decision Letter · Decision Letter 0]

13 Nov 2019

PONE-D-19-21810

How much synthetic oxytocin is infused during labour? A review and analysis of regimens used in 12 countries

PLOS ONE

Dear Authors,

Thank you for submitting your manuscript to PLOS ONE. After careful consideration, we feel that it has merit but does not fully meet PLOS ONE’s publication criteria as it currently stands. Therefore, we invite you to submit a revised version of the manuscript that addresses the points raised during the review process.

We would appreciate receiving your revised manuscript by Dec 22 2019 11:59PM. To enhance the reproducibility of your results, we recommend that if applicable you deposit your laboratory protocols in protocols.io, where a protocol can be assigned its own identifier (DOI) such that it can be cited independently in the future. For instructions see: http://journals.plos.org/plosone/s/submission-guidelines#loc-laboratory-protocols

We look forward to receiving your revised manuscript.

Kind regards,

Salvatore Andrea Mastrolia, M.D.

Academic Editor

PLOS ONE

Journal requirements;

1. Please include additional information regarding the survey or questionnaire used in the study and ensure that you have provided sufficient details that others could replicate the analyses. For instance, please give more details on how the survey was distributed, hoe many participants per each country were invited (and the inclusion and exclusion criteria used), how participating countries were chosen.

Moreover, if you developed a questionnaire as part of this study and it is not under a copyright more restrictive than CC-BY, please include a copy, in both the original language and English, as Supporting Information. Moreover, please include more details on how the questionnaire was pre-tested, and whether it was validated.

2. Please provide additional details regarding participant consent. In the ethics statement in the Methods and online submission information, please ensure that you have specified (1) whether consent was informed and (2) what type you obtained (for instance, written or verbal). If your study included minors, state whether you obtained consent from parents or guardians. If the need for consent was waived by the ethics committee, please include this information.

Reviewers' comments:

Reviewer's Responses to Questions

**Comments to the Author**

1. Is the manuscript technically sound, and do the data support the conclusions?

Reviewer #1: Yes

2. Has the statistical analysis been performed appropriately and rigorously? 

Reviewer #1: Yes

3. Have the authors made all data underlying the findings in their manuscript fully available?

Reviewer #1: Yes

4. Is the manuscript presented in an intelligible fashion and written in standard English?

Reviewer #1: Yes

5. Review Comments to the Author

Reviewer #1: Interesting and well-designed study. Needs minor editing for compound sentences and minor grammatical errors, such as changing "exceeded" to "exceeds" from lines 188-193.

Tables are very detailed, but I think Table 3 is actually too much detail; consider changing this to brief paragraph summaries of different countries' criteria. Similarly- The second part of Table 5, demonstrating all of the possible dosing per each algorithm, detracts from the impact of the work rather than adding to it. I realize this table is the bulk of the outcome of your carefully crafted study, but it is unlikely that a reader would carefully analyze the math. Perhaps including one or two country's data, with an appendix or link out to the full data set? The example at line 404 is terrific and could be bolstered by adding a contrasting example. The big picture differences in initial dose, escalation protocols, and total dose are the staggering outcomes, and you have laid these out well without needing such detail.

The multitude of recent studies and meta-analysis regarding induction at 39 weeks and improved maternal and neonatal outcomes would argue against your tenet of over-medicalizing labor, but looking at the association of bad outcomes with possible injudicious use of exogenous oxytocin remains very important- did you look at any institutional data on cesarean rate, uterine rupture rate, or any other discrete poor outcomes of the differing protocols? Clearly could be a robust additional study but would entice the reader to think about the implications of these differences.

6. PLOS authors have the option to publish the peer review history of their article (what does this mean?). If published, this will include your full peer review and any attached files.

Reviewer #1: Yes: Margaret Larkey Dow

---

## [Author Response · Author response to Decision Letter 0]

11 Dec 2019

We have uploaded a 'Response to reviewers' document

---

## [Decision Letter · Decision Letter 1]

6 Jan 2020

How much synthetic oxytocin is infused during labour? A review and analysis of regimens used in 12 countries

PONE-D-19-21810R1

Dear Authors,

We are pleased to inform you that your manuscript has been judged scientifically suitable for publication and will be formally accepted for publication once it complies with all outstanding technical requirements.

With kind regards,

Salvatore Andrea Mastrolia, M.D.

Academic Editor

PLOS ONE

Reviewers' comments:

Reviewer's Responses to Questions

**Comments to the Author**

1. If the authors have adequately addressed your comments raised in a previous round of review and you feel that this manuscript is now acceptable for publication, you may indicate that here to bypass the “Comments to the Author” section, enter your conflict of interest statement in the “Confidential to Editor” section, and submit your "Accept" recommendation.

Reviewer #1: All comments have been addressed

2. Is the manuscript technically sound, and do the data support the conclusions?

Reviewer #1: (No Response)

3. Has the statistical analysis been performed appropriately and rigorously? 

Reviewer #1: (No Response)

4. Have the authors made all data underlying the findings in their manuscript fully available?

Reviewer #1: (No Response)

5. Is the manuscript presented in an intelligible fashion and written in standard English?

Reviewer #1: (No Response)

6. Review Comments to the Author

Reviewer #1: This study highlights and makes efforts to quantify the considerable variance in synthetic oxytocin administration in several countries, extrapolating timing to max doses and multiple other clinically relevant issues with shocking variability, particularly given its nearly ubiquitous use on labor units. It was an ambitious undertaking, but in its original form it was difficult to appreciate the major findings. The authors took great care to incorporate previous feedback and have added very high impact graphics (notable Figures 1 and 2). Multiple tables were carefully redone, highlighting important data and making them digestible. The manuscript also contains some significant reworking that emphasizes important findings, no longer getting bogged down in a sea of data. This is an important study that is significantly more readable and more powerful in its rewritten form. It is likely to generate much needed renewed interest in exacting appropriate use of this drug.

7. PLOS authors have the option to publish the peer review history of their article (what does this mean?). If published, this will include your full peer review and any attached files.

Reviewer #1: Yes: Margaret Dow

---

## [Editor Report · Acceptance letter]

8 Jul 2020

PONE-D-19-21810R1 

How much synthetic oxytocin is infused during labour? A review and analysis of regimens used in 12 countries 

Dear Dr. Daly:

I'm pleased to inform you that your manuscript has been deemed suitable for publication in PLOS ONE. Congratulations! Your manuscript is now with our production department. 

Kind regards, 

on behalf of

Dr. Salvatore Andrea Mastrolia 

Academic Editor

PLOS ONE